# Production of a Novel Tetrahydroxynaphthalene (THN) Derivative from *Nocardia* sp. CS682 by Metabolic Engineering and Its Bioactivities

**DOI:** 10.3390/molecules24020244

**Published:** 2019-01-10

**Authors:** Ravindra Mishra, Dipesh Dhakal, Jang Mi Han, Haet Nim Lim, Hye Jin Jung, Tokutaro Yamaguchi, Jae Kyung Sohng

**Affiliations:** 1Institute of Biomolecule Reconstruction (iBR), Department of Life Science and Biochemical Engineering, Sun Moon University, 70 Sun Moon-ro 221, Tangjeong-myeon, Asan-si, Chungnam 31460, Korea; ravindra.mishra44@gmail.com (R.M.); medipesh@gmail.com (D.D.); gkswkdal200@naver.com (J.M.H.); gotala9210@naver.com (H.N.L.); poka96@sunmoon.ac.kr (H.J.J.); yamaguchi@sunmoon.ac.kr (T.Y.); 2Department of BT-Convergent Pharmaceutical Engineering, Sun Moon University, 70 Sun Moon-ro 221, Tangjeong-myeon, Asan-si, Chungnam 31460, Korea

**Keywords:** *Nocardia* sp. CS682, Type III Polyketide Synthase, tetrahydroxynaphthalene biosynthesis, UV-protection

## Abstract

Nargenicin A1 is major secondary metabolite produced by *Nocardia* sp. CS682, with an effective antibacterial activity against various Gram-positive bacteria. Most *Nocardia* spp. have metabolic ability to produce compounds of diverse nature, so one-strain-many-compounds (OSMAC) approach can be applied for obtaining versatile compounds from these strains. In this study, we characterized a novel 1, 3, 6, 8-tetrahydroxynaphthalene (THN) derivative by metabolic engineering approach leading to the inactivation of nargenicin A1 biosynthesis. By using genome mining, metabolite profiling, and bioinformatics, the biosynthetic gene cluster and biosynthetic mechanism were elucidated. Further, the antibacterial, anticancer, melanin formation, and UV protective properties for isolated THN compound were performed. The compound did not exhibit significant antibacterial and cytotoxic activities, but it exhibited promising UV protection effects. Thus, metabolic engineering is an effective strategy for discovering novel bioactive molecules.

## 1. Introduction

Natural products (NP)s from microorganisms, plants, and higher organisms are important sources of antimicrobials and anti-tumor agents used in drug development [1,2,3]. Actinobacteria are an important group of microorganisms that are importantly associated with the production of several bioactive compounds [4]. Above all *Streptomyces* species have been reported as the major producers of such biomolecules [5]. Rare-actinobacteria commonly categorized as “non-*Streptomyces*” are less frequently isolated from normal habitats. Recently, such rare-actinobacteria have drawn huge attraction in the search of novel and effective drug leads [2,6]. *Nocardia* species are rare-actinobacteria with promising ability to produce diverse bioactive molecules [7,8]. These molecules possess important bioactivities such as antibacterial [9,10], cytotoxic [11], and immuno-suppressing activities [12]. 

Most of the bioactive molecules are produced by large enzyme complexes often termed as non-ribosomal peptide synthetases (NRPSs) and polyketide synthases (PKSs) [13]. Bacterial PKSs belong to three categories: type I PKS, type II PKS and type III PKS. Most of the type I PKS consists of multifunctional, multi-modular enzymes with non-iterative actions responsible for the catalysis of one cycle of polyketide chain elongation. Despite the “canonical” organization and co-linearity in most of type I PKSs, “non-canonical” examples of type I PKS are continually emerging [14,15]. Type II PKSs constitute multienzyme complexes with a single set of iteratively acting activities [16]. Type III PKSs are generally homo-dimeric enzymes that are iteratively acting condensing enzymes [17]. Type I and II PKSs generally use acyl carrier protein (ACP) to activate acyl-CoA substrates, and channel the growing polyketide intermediates. In case of type III PKSs, the biosynthesis is made through direct action of enzyme on acyl-CoA substrates, which is independent of ACP [14,15,16]. Despite structural and mechanistic differences, all types of PKS compounds are biosynthesized by sequential decarboxylative condensation of acyl-CoA precursors. The most crucial step in the biosynthesis of such compounds is the formation of C–C bonds present in the molecular backbone that is catalyzed by a ketoacyl synthase (KS) domain in case of PKS type I, and subunit architecture in the case of type II and III PKSs [16]. Thus, type III PKS alone is responsible for the formation of C–C bonds through a complete series of decarboxylation, condensation, and cyclization reactions confined with a single active site [18]. The diversity of such type III PKS is determined by the selectivity of starter and extender units, number of condensations, and intramolecular cyclization patterns [19].

Depending on the structures of the products, bacterial type III PKSs are categorized into major five groups: PhlD from *Pseudomonas fluorescens* generating phloroglucinol, DpgA from *Amycolatopsis orientalis* producing 3,5-dihydroxyphenylglycine (DHPG), SrsA from *Streptomyces griseus* generating alkylresorcinols and alkylpyrones, germicidin synthase from *Streptomyces coelicolor* generating alkylpyrones as germicidin, and RppA from various actinomycetes generating 1,3,6,8-tetrahydroxynaphthalene (THN) [18]. The chemical structures of major bacterial type III PKS products are represented in Figure 1. 

THNs have been characterized as the major metabolite of type III PKSs from several actinomycetes species, including *Saccharopolyspora erythrea* [20], *Streptomyces griseus* [21], *S. coelicolor* [22], *Streptomyces peucetius* [23], *Streptomyces toxytricini* [24], and *Sorangium cellulosum* [25]. In most cases, THN is auto-oxidized to form flaviolin, that may occur in monomer or multimer forms [26]. Generally, the oxidation of hydroxyl group is hypothesized to be catalyzed by homologues of *rppB*, which is oxidizing gene present in different strains [20,26]. Thus different flaviolin derivatives are generated by oxidative coupling to form dimers and trimers [24,26]. Similarly, different flaviolin derivatives decorated with rhamnose [20] or prenylated [27] have been reported. Genome mining approaches mediated by genome sequencing, bioinformatics and metabolite profiling are an effective approach for associating the bacterial gene cluster (BGCs) with particular bioactive compounds [2,28,29,30]. However, most of these gene clusters are generally cryptic under normal lab conditions [31]. Gene knock out and comparative metabolic profiling have been used as popular methods for genome mining to identify products of such cryptic BGCs [32]. Such precise metabolic engineering and genome mining approaches are popular for discovering novel secondary metabolites. The identification of cryptic NPs by inactivation/mutation of a biosynthetic gene cluster of a primary product has been utilized to characterize diverse compounds such as nocardamine from *Streptomyces atratus* [33] and geosmin from *S. peucetius* [34]. 

Nargenicin A1, with promising antibacterial activity against various Gram positive bacteria including methicillin resistant *Staphylococcus aureus* (MRSA), is major secondary metabolite produced by *Nocardia* sp. CS682 [35,36]. Most members of rare actinobacteria including *Nocardia* spp. have metabolic ability to produce compounds of diverse nature. Therefore, a one-strain-many-compounds (OSMAC) approach can be applied for activating different metabolic pathways to obtain these versatile compounds. In this study, we were able to activate the production, and characterize novel THN compound by inactivation of the production of nargenicin A1 in *Nocardia* sp. CS682. The structure of the novel compound was determined by mass spectrometric and NMR spectroscopic analyses. The genome mining approach was used to connect the genome information with its BGC. The biosynthetic mechanism of this novel compound was also predicted. The biological importance of this compound was assessed by evaluating its antibacterial, anticancer, melanin formation, and UV-protectant activities. 

## 2. Results and Discussion

### 2.1. Construction of Nargenicin Deletion Mutant

Production of secondary metabolites (SM) can be overlooked due to low production levels, large metabolic background, or unsuitable culture condition [37]. Hence, to enable the production of potential new secondary metabolites from *Nocardia* sp. CS682, a deletion mutant *Nocardia* sp. CS682DR was generated by deletion of the entire PKS region of nargenicin. The deletion mutant generated by double cross-over recombination was confirmed by polymerase chain reaction (PCR) analysis by probes designed for amplification of internal region of nargenicin PKS. There was PCR product in the case of wild type, *Nocardia* sp. CS682, whereas no PCR products were observed in case of *Nocardia* sp. CS682DR (Appendix A). Further PCR of *tsr^r^* (thiostrepton resistance gene used as marker gene for the selection of deletion mutant) was done. There was PCR product in case of the deletion mutant but no product in case of wild type, *Nocardia* sp. CS682 (Appendix A). The loss of the vector (deletion vector) was confirmed by PCR of the apramycin resistance gene present in the vector (*apr^r^*). There was no PCR product in case of both the wild type and the deletion mutant, whereas there was PCR product in case of the pULVK2A (positive control) (Appendix A). The confirmed deletion mutant was named as *Nocardia* sp. CS682DR. Thus, *Nocardia* sp. CS682DR could serve as an excellent starting strain for the characterization of other secondary metabolites produced by *Nocardia* sp. CS682.

In most cases, the morphology, biochemical features, and growth pattern of actinomycetes are influenced by gene deletion. Observation of morphological features showed that there was no significant change, although *Nocardia* sp. CS682DR showed reduced growth rate compared to the wild type strain *Nocardia* sp. CS682. However, ultra-pressure liquid chromatography (UPLC)-photodiode array (PDA) analysis showed that there was significant change in the metabolite profile of *Nocardia* sp. CS682DR compared to that of *Nocardia* sp. CS682. Presence of peak 1 (retention time (t_R_): 5.20 min) was observed in *Nocardia* sp. CS682 as major metabolite, however, peak 1 was absent in the deletion mutant, *Nocardia* sp. CS682DR (Figure 2). Notably, peak 3 (t_R_: 4.81 min) was prominent in *Nocardia* sp. CS682DR (Figure 2). 

### 2.2. Fermentation, Isolation, and Structural Elucidation of the Compound IBR-3 from Nocardia sp. CS682DR

For the isolation and structural elucidation of compound corresponding to peak 3 (Figure 2), fermentation of *Nocardia* sp. CS682DR was carried out in 4L DD media. After several rounds of preparatory-high pressure liquid chromatography (prep-HPLC), 30 mg of purified compound was obtained. The interpretation by ^1^H (Table 1 and Appendix A ), ^13^C NMR (Table 1 and Appendix A), correlation spectroscopy (COSY) (Appendix A), rotational-frame NOE spectroscopy (ROESY) (Appendix A), heteronuclear single quantum correlation (HSQC) (Appendix A), and heteronuclear multiple bond correlation (HMBC) (Appendix A) revealed the structure corresponding to compound IBR-3. By these analyses, the new compound was identified as 1-(α-l-(2-*O*-methyl)-6-deoxymannopyranosyloxy)-3,6,8-trimethoxynaphthalene. 

### 2.3. Metabolite Profiling and Mass Spectrometric Analysis

The extracts from *Nocardia* sp. CS682 and *Nocardia* sp. CS682DR were further analyzed by mass spectrometry. High-resolution quadrupole-time of flight electrospray ionization-mass spectrometry (HR-QTOF ESI/MS) analysis showed that the compound corresponding to the peak 1 in *Nocardia* sp. CS682 was nargenicin ([M+Na]^+^
*m*/*z*^+^ calculated exact mass for C_28_H_37_NNaO_8_^+^, 538.2417, observed mass 538.2428) (Appendix A). However, the UPLC analysis of extract from *Nocardia* sp. CS682DR revealed that peak 2, peak 3, peak 4, and peak 5, were different from those of nargenicin A1 (Figure 2). Further analysis of peak 2, peak 3, peak 4 and peak 5 by mass analysis predicted the production of different THN derivatives corresponding to different peaks at different t_R_, as represented in Table 2 and Appendix A.

### 2.4. Bioinformatic Analysis of Novel THN Gene Cluster and Proposed Biosynthetic Pathway

Whole genome sequencing of *Nocardia* sp. CS682 was performed. Rapid annotation using subsystem technology (RAST) [38] was used for analysis of genome to determine the genes involved in the biosynthesis of THN. In its genome, we could only find a single type III PKS gene bordering with glycosyltransferase. To our knowledge, this is the first report of THN derivatives in which hydroxyl groups are not further oxidized. In addition, the attachment of a methyl group and/or 2-*O*-methyl-rhamnose (2-*O*-methyl-6-deoxymannose) at various hydroxyl positions is also unique. The absence of additional oxidizing enzymes as CYP450 and monooxygenase in the gene cluster can be most possible reason for it. The characteristic presence of methyltransferase and glycosyltransferase genes bordering the PKS gene may be responsible for post-modifications of THN. Hence, based on the analysis of genomic sequence and the compound structure, the biosynthetic gene cluster for novel THN was predicted (Table 3). 

Similarly, the biosynthetic pathway was hypothesized as shown in Figure 3, based on the profile of the confirmed biosynthetic intermediates during metabolite analysis. The predicted ~13kb gene cluster included different genes responsible for formation of the core THN moiety and its post modification (Figure 3). The core THN scaffold is formed by the putative type III PKS synthase *thnA*, which is type III PKS belonging to 1,3,6,8-tetrahydroxynaphthalene synthase (THNS). THN can be further methylated by methyltransferase enzymes *thnM1* or *thnM2* or *thnM3* to form compound IBR-1. The compound IBR-1 is rhamnosylated by glycosyltransferase *thnG* to form compound IBR-2. However, there is no genes involved in biosynthesis of (nucleotide diphosphate sugars) NDP-sugars corresponding to approximately 20 kb upstream and downstream of putative biosynthetic gene cluster. In structurally related compounds like 7-*O*-rhamnosyl flaviolin, the NDP sugar is proposed to be derived from secondary metabolite biosynthetic pathway or borrowed from cell wall biosynthesis, since TDP-L-rhamnose is present in significant amounts as a substrate [20]. In some secondary metabolite biosynthetic pathways like those of elloramycin [39], spinosyn [40] and doxorubicin [41], there is sharing of biosynthetic genes located at different chromosomal loci which may be involved in the biosynthesis of secondary metabolites or cell wall biosynthesis. 

In the case of caprazamycin biosynthesis, it has been reported that there is distantly located sub-cluster for l-rhamnose [42]. Therefore, in the case of formation of compound IBR-2, the biosynthesis of NDP-sugars may be derived from a distantly located biosynthetic gene cluster or a distantly located sugar biosynthetic sub-cluster or cell wall biosynthetic enzymes. Compound IBR-1 and IBR-2 can be further methylated at the sugar moiety by methyltransferase *thnM1*, *thnM2* or *thnM3* to generate compound IBR-3 and compound IBR-4 respectively. The major facilitator superfamily (MFS) proteins, *thnT1* and *thnT2* can be involved as transporter proteins responsible for outflux of secondary metabolites from the cell [43]. The DNA sequences determined in as biosynthetic gene cluster of compounds IBR-1, IBR-2, IBR-3 and IBR-4 from *Nocradia* sp. CS682 in this study have been deposited in the GeneBank database with accession number MK153302.

Further, the phylogenetic analysis was performed for *thnA*, by using the Clustal Omega program, and the calculated results were visualized as unrooted trees by the TreeView program, as shown in Figure 4. The results showed that 1, 3, 6, 8-tetrahydroxynaphthalene synthase (THNS), *thnA* (*Nocardia* sp. CS682), Q9FCA7 (*S. coelicolor* A3(2)) and Q54240 (*S. griseus*) existed in the same clade, which was different from other characterized type III PKS from various actinomycetes.

### 2.5. Analysis of Acyl-CoAs and Transcript Level

The comparison of the titer of acyl-CoAs in wild type and mutant strain indicated that there was significant enhancement in titer of both acyl-CoAs. There was ~5.8 higher titer of acetyl-CoA and ~9.6 higher titer of malonyl-CoA in *Nocardia* sp. CS682DR compared to *Nocardia* sp. CS682 (Table 4). It has been observed that nargenicin A1 utilizes five acetate building blocks. The acetate is converted to malonyl-CoA and incorporated in the molecule [44]. Thus, loss of production of nargenicin leads to accumulation of the short chain fatty acid precursors. The expression levels of biosynthetic genes are crucial for determining the production of a particular metabolite. Hence, the discrepancy in the expression level of the biosynthetic genes of compound IBR-3 in *Nocardia* sp. CS682DR in comparison to *Nocardia* sp. CS682 was assessed. The results of qPCR showed that there was significant enhancement in transcript level of each putative biosynthetic genes (Figure 5) compared to bordering genes, *orf1–2* and *orf3*. This strongly supports the conclusion that the increased transcript levels of each biosynthetic genes may be involved in the enhanced production of compound IBR-3 and its biosynthetic intermediates. Thus, it is imperative to assume that both the availability of acyl-CoAs and the elevated level of expression of biosynthetic genes are key governing factors for the enhanced production of compound IBR-3 and its intermediates from *Nocardia* sp. CS682DR. 

### 2.6. Bioactivities of Novel THN Derivative

Assessment of antibacterial activities indicated that the novel THN derivative did not exhibit significant antimicrobial activity against either Gram-positive or Gram-negative bacteria. There was no clear zone even at concentration of 200 µg/mL and minimum inhibitory concentration (MIC) was greater than 1 mg/mL. The THN derived compounds as fluraquinocin A and neo-marinone exhibit potent antitumor or cytotoxic activity [45,46], so the anticancer potential of compound IBR-3 was evaluated against different cell lines but no cytotoxicity was observed against the most of them (Appendix A). THN is a precursor to the formation of melanin in *S. griseus* [21]. Melanin has various biological functions as it can act as a powerful cation chelator and a free radical sink [47]. This novel THN derivative, compound IBR-3 did not inhibit α-Melanocyte-stimulating hormone (MSH)-induced melanin formation, although it increased melanin formation (Appendix A). The molecules derived from THN such as melanin and flaviolin are reported to exhibit UV protection activities [23,24]. In our experiment there was significant increase in survival rates of cells irradiated with UV in the presence of compound IBR-3. There was positive correlation between the survival rates of cell with increasing concentration of the compounds being evaluated (Figure 6). The anti-photo-ageing and photo-protective compounds particularly effective against UV light are valuable sources for the development of cosmeceutical and topical pharmaceutical products [48]. Thus, compound IBR-3 can be a promising molecule with potential application in cosmetics for skin care and protection.

## 3. Materials and Methods

### 3.1. General Information

All bacteria, plasmids, and primers used in study are listed in Appendix A. Antibiotics were purchased from Sigma Aldrich Chemical Company (St. Louis, MO, USA). Isopropyl-β-d-thiogalactopyranoside (IPTG) was obtained from GeneChem Inc. (Daejeon, Korea). All other chemicals were of the highest grade commercially available. PCR polymerase and related reagents were purchased from Takara Bio Inc. (Shiga, Japan). Cloning vector pGEM®-T Easy was procured from Promega (Madison, WI, USA). All genetic manipulations of DNA were done in *Escherichia coli* XL1 Blue MRF (Stratagene, San Diego, CA, USA). *E. coli* JM110 was used for demethylation before electroporation. Luria Bertani (LB) media (Difco, Franklin Lakes, NJ, USA) and 37 °C were used for maintaining all *E. coli* strains.

The inactivation of the biosynthetic gene cluster was carried out using pULVK2A (Appendix A) by deletion of entire PKS region of nargenicin A1. The upstream and downstream arm were amplified using the respective primers (Appendix A). The amplified upstream region and downstream region were cloned into subcloning vector pKC1139 using *Bam*HI-*Kpn*I and *Kpn*I-*Hind*III sites respectively to generate pKUDN. The pKUDN was digested with *Kpn*I and thiostrepton resistance gene (*tsr^r^*) was ligated to make the final construct pKDN. The *Bam*HI-*Hind*III fragment was digested from pKDN and ligated to pULVK2A to generate pDN. The final construct pDN was transformed into *E. coli* JM110 for demethylation, and used for inactivation of nargenicin A1 biosynthesis by double cross over homologous recombination.

Culture and genetic manipulation of *Nocardia* sp. CS682 and its recombinant strains were performed following standard protocols [7,49]. Brain heart infusion (BHI) medium (Difco, Franklin Lakes, NJ, USA) was used for seed culture. Dipesh Dhakal (DD) media (media optimized by our group for maximal production of polyketides from *Nocardia* spp.), containing 0.2% oatmeal, 0.2% yeast extract, 0.2% soybean meal, 0.1% calcium chloride, 1% maltose, 0.1% magnesium chloride, and 0.4% glycerol was used as production medium for all strains in this study. Incubation at 37 °C and 200 rpm was used for culture of *Nocardia* sp. CS682 and its deletion mutant. 

HPLC grade trifluoroacetic acid (TFA) and acetonitrile were purchased from Mallinckrodt Baker (Phillipsburg, NJ, USA). Other chemicals were of high-grade quality. They were purchased from commercially available sources. All remaining organic solvents were bought from Daejung Chemicals and metals Co. Ltd. (Shiheung, Korea). Dimethyl sulfoxide-*d_6_* (DMSO-*d_6_*) was purchased from Sigma-Aldrich (St. Louis, MO, USA). 

### 3.2. Inactivation of Nargenicin A1 Biosynthetic Gene Cluster and Comparative Analysis

Nargenicin A1 is a polyketide compound belonging to type I PKS. The PKS assembly line is responsible for the formation of the essential multi-carbon scaffold or aglycon, which in turn is modified by post PKS enzymes. The deletion construct pDN was constructed for deletion of entire PKS region of nargenicin A1 biosynthetic gene cluster, as described previously. The deletion construct pDN was transformed in *Nocardia* sp. CS682, and inactivation mutant was generated by double cross over homologous recombination. 

### 3.3. Fermentation and Isolation of Compound IBR-3

*Nocardia* sp. CS682 and *Nocardia* sp. CS682DR were cultured in 500 mL flasks containing 50 mL BHI media and incubated at 37 °C and 200 rpm for 2 days for seed culture. Approximately 5% seed *v*/*v* was inoculated in a 500 mL flask containing 50 mL DD media, and incubated at 37 °C and 200 rpm for 8 days. After 8 days of culture, the culture broth was centrifuged, and the supernatant was extracted with double volumes of ethyl acetate. The extract was dried using a rotary evaporator and concentrated with methanol. The extract was then analyzed by UPLC to obtain peaks corresponding to different SMs. UPLC-PDA analysis was performed using a reverse-phase C_18_ column (ACQUITY UPLC BEH, C_18_, 1.7 μm, Milford, MA, USA) connected to a PDA (UPLC-LG 500 nm) at an absorbance of 254 nm. The HR-QTOF ESI/MS analysis was performed in positive ion mode using SYNAPTG2-S (Waters Corp., Milford, MA, USA). The target compound was purified by prep-HPLC with C18 column (YMC-Pack ODS-AQ-HG, 250 × 20 mm, 10 µm, Shimogyo-ku, Kyoto, Japan) connected to UV detector (254 nm) using 46 min binary gradient at flow rate of 1 mL/min. The solvent system used in both prep-HPLC and UPLC included water (0.025% trifluoroacetic acid) and 100% acetonitrile. However, in UPLC, flow rate was maintained at 0.3 µL/min for 12 min, whereas the acetonitrile concentrations were adjusted as: 0–100% from 0–7 min, 100% from 7–9.50 min, and 100–0% from 9.51–12 min. For determination of structure, the samples were injected in prep-HPLC, whereas there was variation in acetronitrile concentration as: 0–5 min (10%), 5–15 min (30%), 15–25 min (60%), 25–35 min (80%), 35–40 min (95%), 40–45 (10%) and finally stop at 46 min with flow rate of 10%. The peak for IBR-3 corresponding to t_R_: 28.40 min in prep-HPLC was collected. The preparation and processing of the sample for NMR followed a previous protocol [35]. Finally, the freeze-dried sample was dissolved in 600 μL dimethyl sulfoxide-*d6* (DMSO-*d6*), and characterized using a 700-MHz using Avance II 900 Bruker BioSpin NMR spectrometer equipped with a TCI CryoProbe (5 mm, Bruker, Billerica, MA, USA). One-dimensional NMR as ^1^H NMR and ^13^C NMR, and two-dimensional NMR analysis as COSY, ROESY, HSQC, and HMBC were also performed for elucidating the structure of compound IBR-3.

### 3.4. Bioinformatic Analysis 

Preliminarily, RAST analysis of the whole genome was performed to depict the distribution of various gene clusters throughout the genome of *Nocardia* sp.CS682. Type III PKS responsible for the novel THN derivative was identified. Further analysis was performed with BLAST to determine the putative biosynthetic enzymes, which may be involved in the biosynthesis of novel THN derivatives. The putative biosynthetic gene cluster was deposited in National Center for Biotechnology (NCBI). Clustal Omega [50] providing multiple-sequence alignment and a phylogenetic tree based on the neighbor-joining method was used for the analysis of THNS; the phylogenetic tree was represented using TreeView [51]. 

### 3.5. Extraction and Analysis of Acyl-CoAs and Transcript Level 

The loss of nargenicin A1, a polyketide, can affect the short chain CoA titer, which are precursors for polyketide biosynthesis. Hence, concentrations of acetyl-CoA and malonyl-CoA were evaluated according to a previous protocol [35], with slight modification. Silicone oil mixture (800 μL, AR200:DC200, 2:1, δ = 1.01) was added to 500 μL of 15% (*w*/*v*) TCA in a microcentrifuge tube kept on ice. Cell suspension (800 μL) from the culture broth was carefully added to the silicone oil layer without disturbing it. The tube was centrifuged at 20,000× *g* for 5 min at 4 °C, and 300 μL of the TCA extract was immediately drawn using a Pasteur pipette. After extraction and elution from OASIS HLB-SPE cartridge, the elutant was dried in vacuum and reconstituted to 100 µL in water. The sample (20 µL) was then analyzed with a TSKgel amide-80 HPLC column (5 µm diameter, Tosoh Corporation, Tokyo, Japan) using (100 mM potassium phosphate buffer (pH 7.4) and methanol) (95:5 *v*/*v*) as mobile phase for 30 min at 1 mL/min. UV detection (254 nm) was used and the authentic acetyl-CoA and malonyl-CoA were used as references. All the experiments were performed in three replicates and amounts of acetyl-CoA and malonyl-CoA were reported as average with standard error.

For the analysis of transcript level, the RNA was isolated, quantified by nanodrop and purity was determined as described previously [35,49]. The real time PCR(qPCR) was performed using previous protocols [52]. The cDNA was prepared using a High-capacity cDNA Reverse Transcription Kit (Applied Biosystems, Foster City, CA, USA), and equal concentration of template cDNA was used for q-PCR. The qRT-PCR was performed with a QuantiTech SYBER Green RT-PCR Kit (Qiagen, Venlo, The Netherlands). The primers used for qRT-PCR analysis are listed in Appendix A. The primer for 16SrDNA was used as a positive internal control. The relative quantification of the transcriptional level was performed using the 2^−ΔΔCT^ method [53]. Three separate readings were used to represent the final Cf value and the variation was reported as standard deviations. 

### 3.6. Evaluation of Biological Activities 

Antibacterial activity was evaluated against five human pathogens (viz three Gram-positive bacteria, *Bacillus subtilis* KACC 17047, *Staphylococcus aureus* subsp. *aureus* KCTC 1916, *Micrococcus luteus* KACC 13377, and two Gram-negative bacteria *Enterobacter cloaceae* subsp. *disolvens* KACC 13002 and *Pseudomonas aeruginosa* KACC 10232) according to a previous protocol [54]. Afterwards, microbroth dilution method was used to determine MIC values according to previous protocol [55] following the guidelines provided by clinical and laboratories standards institute (CLSI, Wayne, PA, USA). Similarly, the anticancer activity was assessed against different cancer cell lines such as AGS (gastric cell line), HCT116 (colon cancer cell line), HeLa (uterine cervical cell line) and U87MG (brain cell line) according to a previous protocol [35] using the 3-(4,5-dimethylthiazol-2-yl)-2,5-diphenyltetrazolium bromide (MTT) colorimetric assay. The measurement of melanin content was performed according to a standard protocol [56]. B16F10 cells (5 × 10^4^ cells/well) were treated with various concentrations of compound in presence or absence of α-MSH (10 nM) for 72 h. The cells were washed, lysed and absorbance of lysate was measured at 405 nm using a microplate spectrophotometer (Multiskan® Spectrum, Thermo Scientific, Waltham, MA, USA). The UV protection effect of the compound was measured using a previous protocol with slight modification [24]. The overnight culture of *E. coli* BL21(DE3) maintained in LB broth at 37 °C, was centrifuged at 200 rpm and diluted to final concentration of 10^7^ CFU mL^−1^ in phosphate buffer saline (PBS). The compound was added to adjust the final concentration (200 mM, 100 mM, 50 mM and 25 mM) of the diluted sample. The sample without compound was used as control. All the samples amounted to 1 mL in glass vials were exposed in custom-built UV chamber fitted with a UV lamp at the top to adjust the distance of 15 cm. The cells were exposed to 254 nm UV light for 15, 30, 45, and 60 s, respectively. Each sample was diluted by 100 folds and 100 μL of the sample were spread on an LB plate. The surviving cells formed visible colonies after incubation at 37 °C for 16 h, and colonies were counted. The cell survival ratio was calculated as ratio of colonies that appeared after UV treatment to the colonies in the control sample, without UV irradiation.

## Figures and Tables

**Figure 1 molecules-24-00244-f001:**
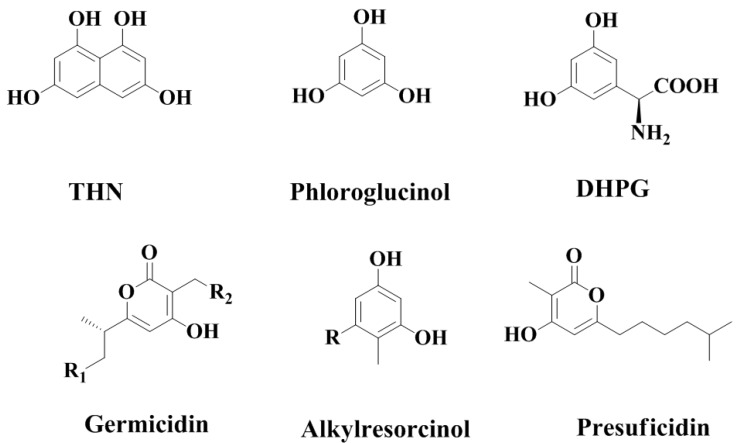
Chemical structures of major bacterial type III polyketide synthase (PKS) products.

**Figure 2 molecules-24-00244-f002:**
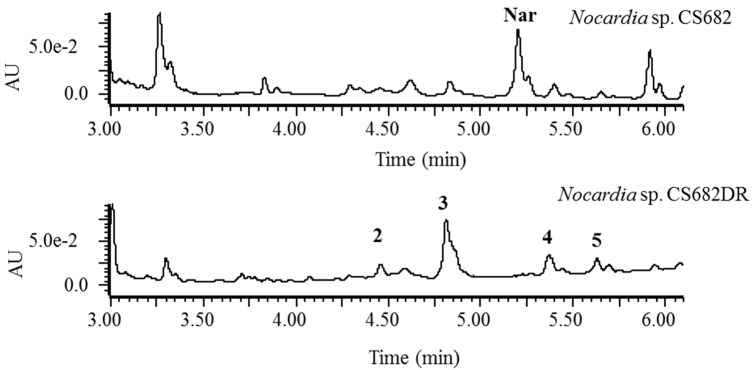
Ultra-pressure liquid chromatography (UPLC) chromatogram of extract from *Nocardia* sp. CS682 and *Nocardia* sp. CS682DR.

**Figure 3 molecules-24-00244-f003:**
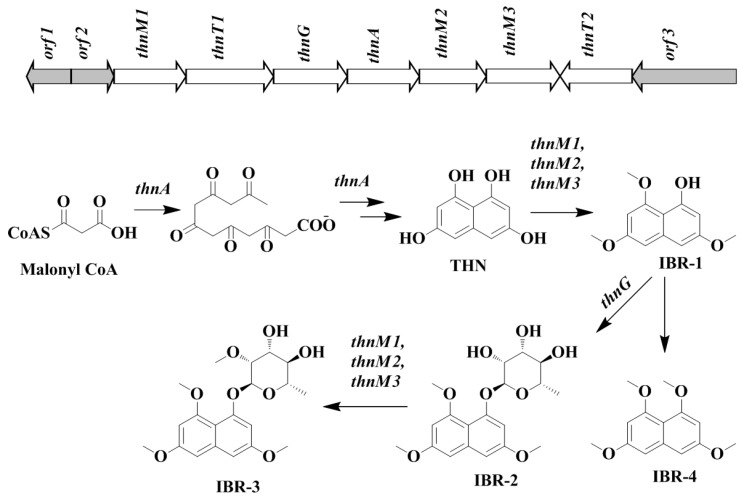
The putative biosynthetic gene cluster and proposed biosynthetic mechanism of compound IBR-3.

**Figure 4 molecules-24-00244-f004:**
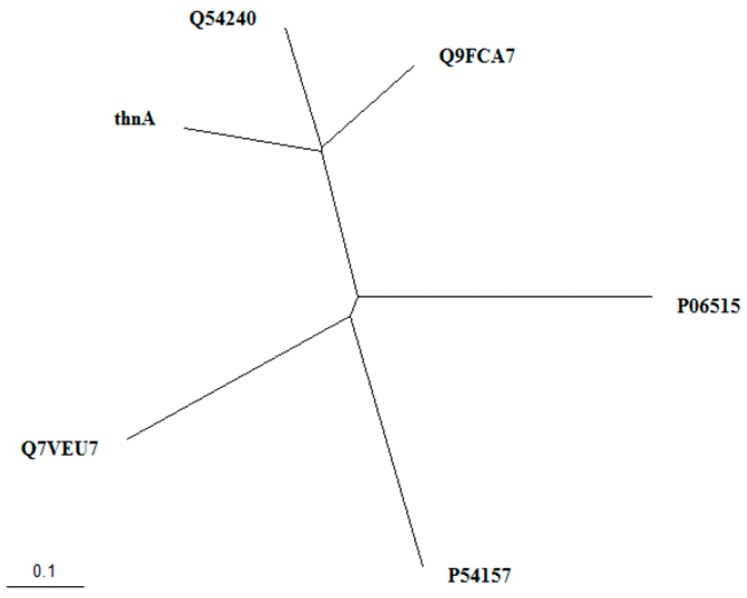
Phylogenetic analysis of bacterial type III PKS related to *thnA* using Clustal Omega and visualized as unrooted trees by TreeView. 1, 3, 6, 8-tetrahydroxynaphthalene synthase (THNS) from different strains, *thnA*: *Nocardia* sp. CS682, Q9FCA7: *Streptomyces coelicolor* A3(2) and Q54240: *Streptomyces griseus*; Chalcone synthase from different strains, P54157: *Bacillus subtilis* subsp. subtilis str. 168 and P06515: *Antirrhinum majus*; Alpha-pyrone synthase, Q7VEU7: *Mycobacterium tuberculosis*.

**Figure 5 molecules-24-00244-f005:**
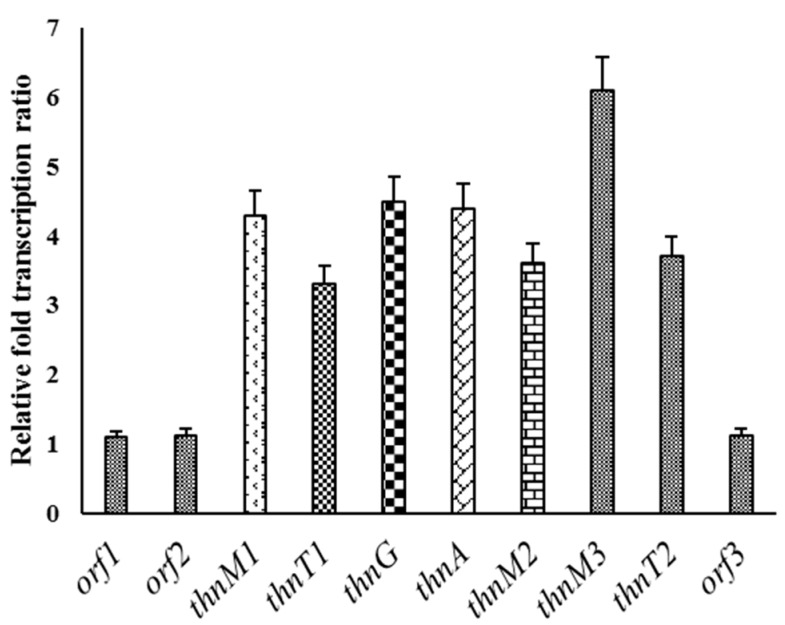
Relative fold increase in the transcript level of different genes in *Nocardia* sp. CS682DR.

**Figure 6 molecules-24-00244-f006:**
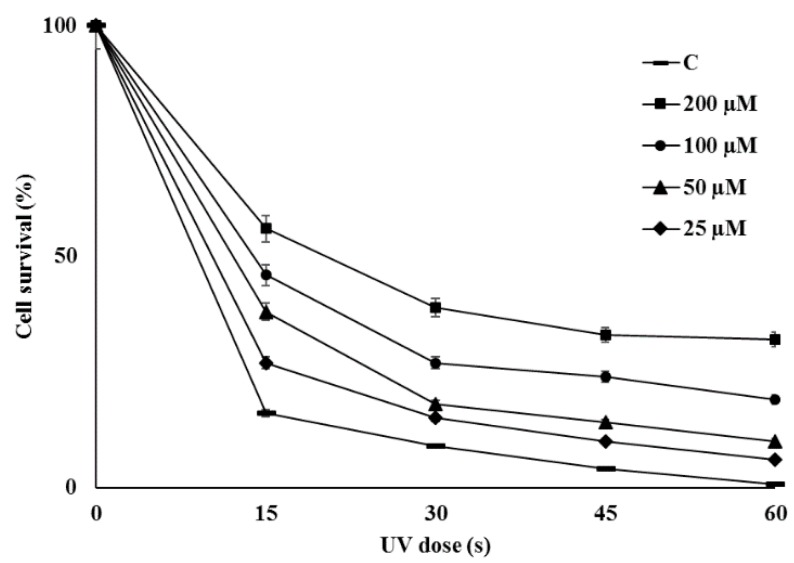
Effect of different concentration of compound IBR-3 on survival rates of UV-irradiated *Escherichia coli* BL21 (DE3).

**Table 1 molecules-24-00244-t001:**
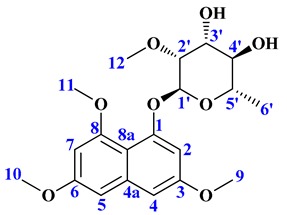
^1^H NMR and ^13^C NMR spectroscopic data (700 MHz) of compound IBR-3 measured in DMSO-*d_6_*^a^.

Position	δc	δ_H_ (*J* in Hz)	Intensities
1′	96.38	5.48 (s)	1H
2′	80.51	3.58 (m)	
3′	70.26	3.88(dd) *J* = 9.6 Hz, 9.6 Hz,	
4′	72.19	3.25(dd) *J* = 9.5 Hz, 9.5 Hz	
5′	69.7	3.58 (m)	
6′	17.92	1.10(d) *J* = 6.2 Hz	3H
1	153.92		
2	102.15	6.56(d) *J* = 2.4 Hz	
3	157.87		
4	100.61	6.87(d) *J* = 2.4 Hz	
5	98.52	6.78(d) *J* = 2.3 Hz	
6	158.32		
7	96.5	6.38(d) *J* = 2.3 Hz	
8	157.56		
9	55.09	3.81(s)	3H
10	55.08	3.82(s)	
11	55.8	3.84(s)	
12	58.73	3.44(s)	3H
4a	138.76		
8a	108.62		

^a^ The respective chemical shifts in ppm are indicated by δ. Multiplicities are indicated by s (singlet), d (doublet), and m (multiplet) including coupling constant *J* in Hz.

**Table 2 molecules-24-00244-t002:** Different compounds predicted from *Nocardia* sp. CS682DR based on mass analysis.

Compound	Name	Structure	Elemental formula	[M + H]^+^ theoretical	[M + H]^+^ observed	Peak in UPLC	t_R_ (min)
IBR-1	3,6,8-trimethoxy naphthalen-1-ol	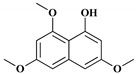	C_13_H_14_O_4_	235.0965	235.0961	4	5.37
IBR-2	1-(α-l-6-deoxy-mannopyranosyloxy)- 3,6,8-trimethoxy naphthalene	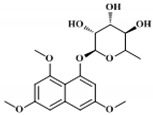	C_19_H_24_O_8_	381.1544	381.1543	2	4.46
IBR-3	1-(α-l-(2-*O*-methyl)-6-deoxymanno- pyranosyloxy)-3,6,8-trimethoxy naphthalene	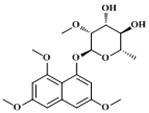	C_20_H_26_O_8_	395.1700	395.1705	3	4.81
IBR-4	1,3,6,8-tetramethoxy naphthalene	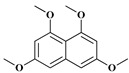	C_14_H_16_O_4_	249.1127	249.1129	5	5.63

**Table 3 molecules-24-00244-t003:** Putative genes involved in the biosynthetic gene cluster of compound IBR-3 ^a^.

Gene	Gene Size (nt)	Putative Function	Best BLAST Hit of the Gene Product		
			Genebank Accession No.	Species	Id/Sim
*orf1*	780	voltage-gated potassium channel	PXX59685	*Nocardia tenerifensis*	68/77
*orf2*	729	hypothetical protein	WP_084160940	*Nocardia* sp. sp. BMG51109	61/71
*thnM1*	1164	hypothetical protein	WP_093574921	*Amycolatopsis rubida*	53/66
		class I SAM-dependent methyltransferase	WP_018807243	*Salinispora arenicola*	52/65
*thnT1*	1293	MFS transporter	WP_107657235	*Nocardia suismassiliense*	99/99
*thnG*	1173	DUF1205 domain-containing protein	WP_033430793	*Saccharothrix syringae*	53/67
		glycosyltransferase	AIW62993	uncultured bacterium BAC-AB1442/1414/561	40/58
*thnA*	1125	type III polyketide synthase	WP_045697370	*Streptomyces rubellomurinus*	68/78
*thnM2*	807	methyltransferase domain-containing protein	WP_084716343	*Saccharothrix syringae*	53/67
*thnM3*	1044	methyltransferase	WP_033430882	*Saccharothrix syringae*	65/75
		Dimerisation domain-containing protein	SDZ58908	*Saccharopolyspora shandongensis*	58/72
*thnT2*	1305	MFS transporter	WP_067522717	*Nocardia uniformis*	87/91
*orf3*	1371	MBL fold metallo-hydrolase	WP_067522716	*Nocardia uniformis*	80/88

^a^ Coverage more than 70% and gene product size >150 amino acids only considered for annotation. nt: nucleotides, orf: open reading frame, Id: identity, Sim: similarity.

**Table 4 molecules-24-00244-t004:** Analysis of acyl-CoA extracted from *Nocardia* sp. CS682 and *Nocardia* sp. CS682DR.

		Acetyl-CoA	Malonyl-CoA
Strains	Day	(µmol/g DCW)	(µmol/g DCW)
*Nocardia* sp. CS682	3	9.24 ± 0.48	3.46 ± 0.12
	6	22.16 ± 5.54	38.84 ± 1.60
*Nocardia* sp. CS682DR	3	24. 64 ± 1.02	18.40 ± 0.96
	6	140.32 ± 2.46	174.78 ± 3.68

DCW: Dry cell weight.

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
