# Peer review of "Production of a Novel Tetrahydroxynaphthalene (THN) Derivative from Nocardia sp. CS682 by Metabolic Engineering and Its Bioactivities"

_molecules, 2019, doi:10.3390/molecules24020244_

Reviewer 1 Report

The manuscript by Mishra et al. describes the identification, characterization and biosynthesis of novel THN derivatives. The authors used a metabolic engineering approach to enhance the production of the compounds. Therefore, they inactivated the production of the major polyketide product in Nocardia sp. which led to the redistribution of the now available acyl-CoA building blocks to other PKS pathways. The science is in general sound, the compounds are novel and the approach is interesting. I have thus no major scientific concerns for this study. However, the authors need to extensively revise their manuscript regarding english grammar, errors, typos, ambiguities, redundancy,  and the overall organization of their text.
Some specific concerns:
-L73-74: "Genome mining is an effective approach for analyzing genomic information and particular bacterial gene clusters associated with particular bioactive compounds". I am not happy with this definition at all. Why is it effective? And why does genome mining help in gene cluster analysis? Isn't that done by certain bioinformatic tools?
-L76-78: Silent, orphan, cryptic gene clusters. Please use the terms correctly.
-L80: What is a silent natural product? Please rephrase.
-L86-87: "However, most of the members of rare actinobacteria including Nocardia spp. have metabolic ability to produce compound of diverse nature in one strain-many compounds  (OSMAC)." Please rephrase.
L104-108: 1. the complete replacement of the WT gene with the resistance gene should be checked by PCR on the WT gene. 2. Please rephrase "There was amplification product in case of Nocardia sp. CS682DR but no amplification in case of Nocardia sp. CS682." and "There was no amplification product in case of both Nocardia sp. CS682 and Nocardia sp. CS682DR, whereas there was amplification product in case of pULVK2A."
L118-128: This part is highly redundant.
Figure 2: please indicate if the compounds are already present in the WT extract.
Figure 2/Table 1/text: the different numbering/numerical abbrevations that are used for the peaks and the correpsonding compounds makes the whole thing very difficult to read. Please revise!
L137-141: Redundant! Please rephrase.
L145: "The whole genome of Nocardia sp. CS682 was performed ...". Please correct.
L154: "However, to our knowledge it is first report of the THN derivatives that are NOT oxidized and decorated with methylation at various hydroxyl positions and/or 2-O methyl-rhamnose (2-O-methyl-6-155 deoxymannose)." I assume the "not" here is an error. Please correct.
L156-159: please rephrase.
L188-189: a similar observation has been made in the caprazamycin pathway by Sohng and others (Appl Environ Microbiol. 2010;76(12):4008-18.) It should be possible to find the cluster elsewhere on the genome of Nocardia sp. CS682. Please check.
L196-197: Why is the gene cluster only for Compound 3? What about 1,2,4?
L205: please rephrase
L211: please correct number of figure.
L229: please correct µm.
Section 2.5: please only describe those experiments in detail which gave positive results. Thereby the section can be shortened extensively. IMO IC50 of >200 µM against cell lines is no cytoxicity, please revise.
L323: please explain the protocol for the acyl-CoA analysis in more detail.

Author Response

Reviewer #1:

General comments:

The manuscript by Mishra et al. describes the identification, characterization and biosynthesis of novel THN derivatives. The authors used a metabolic engineering approach to enhance the production of the compounds. Therefore, they inactivated the production of the major polyketide product in Nocardia sp. which led to the redistribution of the now available acyl-CoA building blocks to other PKS pathways. The science is in general sound, the compounds are novel and the approach is interesting. I have thus no major scientific concerns for this study. However, the authors need to extensively revise their manuscript regarding english grammar, errors, typos, ambiguities, redundancy,  and the overall organization of their text.

Response: According to reviewers’ suggestions, we have consulted English expert and revised the manuscript. We have also amended the tables, texts and references according to reviewer’s suggestions.

Some specific concerns:

-L73-74: "Genome mining is an effective approach for analyzing genomic information and particular bacterial gene clusters associated with particular bioactive compounds". I am not happy with this definition at all. Why is it effective? And why does genome mining help in gene cluster analysis? Isn't that done by certain bioinformatic tools?

Response: Genome mining refers to deriving various information about the organism based on genome analysis. There has been various approaches for genome mining as genomics and bioinformatics (DOI:10.1093/nar/gkr466), and genomics and metabolomics (DOI: 10.1128/mSystems.00028-15 ). Hence, the correction has been made as “Genome mining approaches mediated by genome sequencing, bioinformatic analysis and metabolite profiling is an effective approach for associating the bacterial gene clusters with particular bioactive compounds”

-L76-78: Silent, orphan, cryptic gene clusters. Please use the terms correctly.

Response: “Cryptic” is most appropriate word, hence we have corrected in appropriate places.

-L80: What is a silent natural product? Please rephrase.

Response: We have corrected to “cryptic”, because it is more appropriate.

-L86-87: "However, most of the members of rare actinobacteria including Nocardia spp. have metabolic ability to produce compound of diverse nature in one strain-many compounds  (OSMAC)." Please rephrase.

Response:  The appropriate correction has been made.

L104-108: 1. the complete replacement of the WT gene with the resistance gene should be checked by PCR on the WT gene. 2. Please rephrase "There was amplification product in case of Nocardia sp. CS682DR but no amplification in case of Nocardia sp. CS682." and "There was no amplification product in case of both Nocardia sp. CS682 and Nocardia sp. CS682DR, whereas there was amplification product in case of pULVK2A."

Response: We confirmed the deletion by PCR of different segments of PKS gene, whereas none of them was observed in deletion mutant Nocardia sp. CS682DR but in case of Nocardia sp. CS682 the appropriate PCR products were observed. This data has been supplemented in Fig. S1 and appropriate primer pairs are also included in Table 2.  However, we also confirmed double cross over by PCR of marker genes.

According to reviewer’s suggestion, we have also rephrased the sentence.

L118-128: This part is highly redundant.

Response: The appropriate correction has been duly made.

Figure 2: please indicate if the compounds are already present in the WT extract.

Response: The compounds may be already present in the WT, but most of the peaks also contained other derivatives of nargenicin A1 as prominent compound. So, in mass chromatogram they were not the major compound providing mass fragmentations. In another manuscript, we are revealing about the biosynthetic mechanism of nargenicin A1, which will confirm about the mixture of compounds present in these peaks. So for now, it would not be appropriate to indicate those compounds in WT UPLC profile.

Figure 2/Table 1/text: the different numbering/numerical abbrevations that are used for the peaks and the correpsonding compounds makes the whole thing very difficult to read. Please revise!

Response: The appropriate revision has been duly made.

L137-141: Redundant! Please rephrase.

Response: The correction has been duly made.

L145: "The whole genome of Nocardia sp. CS682 was performed ...". Please correct.

Response: The appropriate correction has been made.

L154: "However, to our knowledge it is first report of the THN derivatives that are NOT oxidized and decorated with methylation at various hydroxyl positions and/or 2-O methyl-rhamnose (2-O-methyl-6-155 deoxymannose)." I assume the "not" here is an error. Please correct.

Response:  The appropriate correction has ben made.

L156-159: please rephrase.

Response: The appropriate correction has been duly made.

L188-189: a similar observation has been made in the caprazamycin pathway by Sohng and others (Appl Environ Microbiol. 2010;76(12):4008-18.) It should be possible to find the cluster elsewhere on the genome of Nocardia sp. CS682. Please check.

Response: We are thankful to reviewer for the suggestion. But in secondary metabolite biosynthetic pathways as elloramycin, spinosyn and doxorubicin there is sharing of biosynthetic genes located at different chromosomal loci which may be involved in biosynthesis of secondary metabolites or cell wall biosynthesis. In case of caprazamycin biosynthesis, it has been reported that there is distantly located sub-cluster for L-rhamnose. Therefore, in the case of formation of compound IBR-2, there is not confirmative information of genes involved in the biosynthesis of NDP-sugar. Hence, we are targeting for gene-inactivation studies in future studies. Therefore, for now we have not included information about the clusters for sugar biosynthesis. However, we have included the reference as suggested by reviewer. 

L196-197: Why is the gene cluster only for Compound 3? What about 1,2,4?

Response: Certainly, it is cluster for compound 1,2,3 and 4. So appropriate correction has been made.

L205: please rephrase

Response: The appropriate correction has been duly made.

L211: please correct number of figure.

Response: The appropriate correction has been made.

L229: please correct µm.

Response: The appropriate correction has been duly made.

Section 2.5: please only describe those experiments in detail which gave positive results. Thereby the section can be shortened extensively. IMO IC50 of >200 µM against cell lines is no cytoxicity, please revise.

Response:  The appropriate correction has been made.

L323: please explain the protocol for the acyl-CoA analysis in more detail.

Response: The protocol has been explained in details

Reviewer 2 Report

The authors describe the deletion of the nargenicin A1 PKS region in the actinomcyes Nocardia sp. CS682 which abolished the production of nargenicin A1. Analysis of the acyl-CoA pool indicated that there was much higher concentrations in the new KO mutant than the WT. UPLC analysis further indicated that there were four new compounds identified in the chemical extract of the KO compared to the WT. Purification and analysis of these compounds by HRMS and NMR identified them as THN-derivatives, one of which is previously unreported. Genome mining identified a putative BGC based on a single Type III PKS clustered with methyltransferase and glycosyltransferase genes and a biosynthesis of the observed compounds is proposed. Biological testing of the novel THN-derivative compound 3 indicated potential applications in cosmetics due to its UV protection activities. 

Overall, I think the research is well designed and presents a useful method of identifying novel compounds that may be overshadowed due to high levels of production of competing molecules which deplete the acyl-CoA building block pool. From a scientific standpoint I think the research is sound. My main comments are related to the presentation / description of the work which I believe can be easily addressed. 

For example, a short conclusion would be nice, although I realise this is optional. From a language point of view one / a novel derivative should be specified to avoid confusion with several novel derivatives which is implied currently. When the authors are describing their chromatography results e.g. the 4 new peaks, they seem to already know, without any further analysis, that they are THN derivatives (Line 120). I would suggest avoiding any reference to these four compounds being THN derivatives until the structure is confirmed by NMR, or unless chemical standards / dereplication database are available. Also unclear is how the structures of compounds 1, 2 and 4 are known. Probably Tables 1 and 2 need to be swapped as results have been presented before analysis. The biosynthetic pathway Figure 4 should actually be Figure 3 as it is mentioned on Line 166, before thy phylogenetic tree which should be Figure 4. 

In the introduction Line 44/45 it is mentioned that Type I PKS are non-iterative. While this is mostly true there are several examples of Type I bacterial PKS being iterative (Review: Chen & Du, Appl. Microbiol. Biotech., 2916, 100, 541-557). Type III PKS are introduced nicely but numbering compounds would help link diagrams to text (e.g. compounds in Figure 1) and the biosynthesis of THN discussed in lines 148-153 might be better placed in the introduction with a short scheme of the whole known biosynthesis. 

Within the experimental some discussion crept in which is unnecessary e.g. Lines 275-281 and 285-287 and 288-291 and 321-322. Reference 20 (Line 248) organism name should be italicised. Formatting of Refs 45 and 46? 

I would recommend this manuscript for publication after these points have been addressed.

Author Response

General comments:

The authors describe the deletion of the nargenicin A1 PKS region in the actinomcyes Nocardia sp. CS682 which abolished the production of nargenicin A1. Analysis of the acyl-CoA pool indicated that there was much higher concentrations in the new KO mutant than the WT. UPLC analysis further indicated that there were four new compounds identified in the chemical extract of the KO compared to the WT. Purification and analysis of these compounds by HRMS and NMR identified them as THN-derivatives, one of which is previously unreported. Genome mining identified a putative BGC based on a single Type III PKS clustered with methyltransferase and glycosyltransferase genes and a biosynthesis of the observed compounds is proposed. Biological testing of the novel THN-derivative compound 3 indicated potential applications in cosmetics due to its UV protection activities.

Overall, I think the research is well designed and presents a useful method of identifying novel compounds that may be overshadowed due to high levels of production of competing molecules which deplete the acyl-CoA building block pool. From a scientific standpoint I think the research is sound. My main comments are related to the presentation / description of the work which I believe can be easily addressed.

Response: According to reviewers’ suggestions, we have consulted English expert and revised the manuscript. We have also amended the tables, texts and references according to reviewer’s suggestions.

Comments: For example, a short conclusion would be nice, although I realise this is optional. From a language point of view one / a novel derivative should be specified to avoid confusion with several novel derivatives which is implied currently. When the authors are describing their chromatography results e.g. the 4 new peaks, they seem to already know, without any further analysis, that they are THN derivatives (Line 120). I would suggest avoiding any reference to these four compounds being THN derivatives until the structure is confirmed by NMR, or unless chemical standards / dereplication database are available. Also unclear is how the structures of compounds 1, 2 and 4 are known. Probably Tables 1 and 2 need to be swapped as results have been presented before analysis. The biosynthetic pathway Figure 4 should actually be Figure 3 as it is mentioned on Line 166, before thy phylogenetic tree which should be Figure 4.

Response:  The appropriate correction has been duly made.

Comments: In the introduction Line 44/45 it is mentioned that Type I PKS are non-iterative. While this is mostly true there are several examples of Type I bacterial PKS being iterative (Review: Chen & Du, Appl. Microbiol. Biotech., 2916, 100, 541-557). Type III PKS are introduced nicely but numbering compounds would help link diagrams to text (e.g. compounds in Figure 1) and the biosynthesis of THN discussed in lines 148-153 might be better placed in the introduction with a short scheme of the whole known biosynthesis. 

Response: The appropriate correction has been duly made.

The reviewer mentions that ” the introduction Line 44/45 it is mentioned that Type I PKS are non-iterative. While this is mostly true there are several examples of Type I bacterial PKS being iterative.”. But we have already quoted the information as “Despite the “canonical” organization and co-linearity in most of type I PKSs, non-canonical examples of type I PKS are continually emerging” and cited  “Chen & Du, Appl. Microbiol. Biotech., 2916, 100, 541-557” as Reference no 14.

The reviewers have suggested numbering compounds would help link diagrams to text (e.g. compounds in Figure 1), but we assume it will cause confusion. Hence, we have included abbreviation in the text for the compounds included in figure 1.

The reviewers have suggested including biosynthesis of THN discussed in lines 148-153 might be better placed in the introduction with a short scheme of the whole known biosynthesis. So, we have made the appropriate correction.

Comments: Within the experimental some discussion crept in which is unnecessary e.g. Lines 275-281 and 285-287 and 288-291 and 321-322. Reference 20 (Line 248) organism name should be italicised. Formatting of Refs 45 and 46?

Response: The appropriate correction has been duly made.
